

# Inconsistent decadal variations between surface and free tropospheric nitrogen oxides over United States

Zhe Jiang[1], Helen Worden[1], John R. Worden[2], Daven K. Henze[3], Dylan B. A. Jones[4], Avelino F. Arellano[5], Emily V. Fischer[6], Liye Zhu[6], Kazuyuki Miyazaki [2,7], K. Folkert Boersma[8,9], Vivienne H. Payne[2],

[1]National Center for Atmospheric Research, Boulder, CO, USA
[2]Jet Propulsion Laboratory, California Institute of Technology, Pasadena, CA, USA
[3]Department of Mechanical Engineering, University of Colorado, Boulder, CO, USA
[4]Department of Physics, University of Toronto, Toronto, ON, Canada
[5]Department of Hydrology and Atmospheric Sciences, University of Arizona, Tucson, AZ, USA
[6]Department of Atmospheric Science, Colorado State University, Fort Collins, CO, USA
[7]Japan Agency for Marine-Earth Science and Technology, Yokohama, Japan
[8]Wageningen University, Meteorological and Air Quality department, Wageningen, the Netherlands
[9]Royal Netherlands Meteorological Institute, De Bilt, The Netherlands



**Abstract**
Decreases in surface emissions of nitrogen oxides ($NO_x = NO + NO_2$) in North America have led
to substantial improvements in air-quality over the last several decades. Here we show that satellite
observations of tropospheric nitrogen dioxide ($NO_2$) columns over the contiguous United States
(US) do not decrease after about 2009, while surface $NO_2$ concentrations continue to decline
through to the present. This divergence, if it continues, could have a substantial impact on surface
air quality due to mixing of free-tropospheric air into the boundary layer. Our results show only
limited contributions from local effects such as fossil fuel emissions, lightning, or instrument
artifacts, but we do find a possible relationship of $NO_2$ changes to decadal climate variability. Our
analysis demonstrates that the intensity of transpacific transport is stronger in El Niño years and
weaker in La Niña years, and consequently, that decadal-scale climate variability impacts the
contribution of Asian emissions on North American atmospheric composition. Because of the short
lifetime, it is usually believed that the direct contribution of long-range transport to tropospheric
$NO_x$ distribution is limited. If our hypothesis about transported Asian emissions is correct, then
this observed divergence between satellite and surface $NO_x$ could indicate mechanisms that allow
for either $NO_x$ or its reservoir species to have a larger than expected effect on North American
tropospheric composition. These results therefore suggest more aircraft and satellite studies to
determine the possible missing processes in our understanding of the long-range transport of
tropospheric $NO_x$.

**1. Introduction**

Nitrogen oxides play a complex role in tropospheric chemistry and have a strong influence

on air quality as precursors in the formation of ozone ($O_3$) and secondary aerosols. Tropospheric



NO$_x$ is produced through anthropogenic combustion, biomass burning, soil (Jaegle et al., 2005),
and lightning emissions (Schumann and Huntrieser, 2007), and is mainly removed by the
formation of nitric acid (HNO$_3$). Most NO$_x$ is emitted as nitric oxide (NO), however, it is most
appropriate to consider the budget of the NO$_x$ as a whole, because of the rapid cycling between
NO and NO$_2$ (~ 1 min). Tropospheric NO$_x$ has short lifetime, a few hours except in extratropical
winter when it increases to 1-2 days (Martin et al. 2003).

Because of the short lifetime, state of the art chemistry/climate models suggest that the

direct contribution of long-range transport to tropospheric NO$_x$ distribution is limited (e.g. Zhang
et al. 2008). However, NO$_x$ can also be transported far away from the sources via the formation of
long-lived reservoir species, such as peroxyacetyl nitrate (PAN, e.g., Fischer et al., 2014; Jiang et
al. 2016a). Models have large uncertainties in PAN abundance (Fischer et al., 2014) and there are
also potentially other missing processes in the chemical transport models used to diagnose NO$_x$
lifetime and transport. For example, a recent discovery about the rapid cycling of reactive nitrogen
in the marine boundary layer (Ye et al. 2016) demonstrates processes that are not represented in
modeled NO$_x$ transport, and that may help explain existing discrepancies in reactive nitrogen
partitioning between models and observations.

Due to its critical influence in the troposphere, there are multiple space-based

measurements for tropospheric NO$_2$ that are available from satellites that were launched in the past
two decades. These instruments typically measure backscattered solar radiation from which the
vertically integrated column abundance of NO$_2$ is retrieved. The assumption of weak long-range
transport allows relatively simple applications of the space-based NO$_2$ column data to study NO$_x$
sources. For example, recent studies (e.g. Reuter et al. 2014; Itahashi et al. 2014; Duncan et al.
2016; Krotkov et al. 2016) assessed the trends of surface NO$_x$ emissions by assuming a strong





correlation between tropospheric $NO_2$ columns with local emissions. The tropospheric $NO_2$
column data are also widely used in inverse modeling analyses to estimate $NO_x$ emissions by either
scaling the surface $NO_x$ emissions with the corresponding ratio of observed over modeled
tropospheric $NO_2$ column (e.g. Lamsal et al. 2011; Mijling et al. 2012; Gu et al. 2014) or through
data assimilation techniques with short localization length scales (e.g. Miyazaki et al. 2017).

Since 1990, US regulations have required significant $NO_x$ emission reductions over many

regions (US Environmental Protection Agency, 2010). The trend of decreasing local US $NO_x$
emissions has been confirmed by several studies (e.g. Lamsal et al. 2015; Tong et al. 2015; Kharol
et al. 2015; Duncan et al. 2016; Krotkov et al. 2016). In contrast to the decreasing local $NO_x$
emissions, recent studies (e.g. Cooper et al. 2010; Verstraeten et al. 2015) have indicated an
increase in free tropospheric $O_3$ over western North America over the past decade. The discrepancy
between variations of local $NO_x$ emissions and free tropospheric $O_3$ suggests possible influences
from non-local sources, and consequently, provides motivation to re-evaluate the contribution of
long-range transport to the free tropospheric $NO_x$ distribution.

In this work, we investigate the variation of US tropospheric $NO_2$ in the past decade to

assess the contribution of non-local sources. We will particularly explore the possible answers for
the following questions: why there is good agreement between tropospheric $NO_2$ column and
surface measurements over the period of 2005-2008? What is the reason for the appearance of the
large and growing divergence at around 2009? What is the impact of the decreasing Chinese $NO_x$
emissions since 2013 (Liu et al. 2016) on North America? To evaluate these critical questions,
multiple data sets and model are used in this work, including remotely sensed $NO_2$ column
measurements from Ozone Monitoring Instrument (OMI, NASA and DOMINO products), in-situ
surface $NO_2$ measurements from the Environmental Protection Agency (EPA) Air Quality System



(AQS) network and the Environment Canada National Air Pollution Surveillance Program (NAPS)
network, flash rate density data from Lightning Imaging Sensor (LIS), and the GEOS-Chem
chemical transport model.
This paper is organized as follows: in Section 2 we describe the observations and model
used in this work. In Section 3 we demonstrate the divergence between the OMI $NO_2$ column
retrievals and surface measurements over the period of 2005-2015 and focus on the evaluation of
contributions from various hypotheses that could explain the divergence. Our conclusions follow
in Section 4.
**2. Observations and Models**
**2.1 Tropospheric $NO_2$ column from OMI**
The OMI instrument was launched on NASA's Aura spacecraft. The sensor has a spatial
resolution of 13 km x 24 km. OMI provides daily global coverage with measurements of both
direct and atmosphere-backscattered sunlight in the ultraviolet-visible range from 270 to 500 nm;
the spectral range 405-465 nm is used to retrieve tropospheric $NO_2$ columns. Two versions of the
OMI retrievals (level 2) are used in this work: the NASA (version 3, Krotkov and Veefkind 2006;
Bucsela et al. 2013) and DOMINO (version 2, Boersma et al. 2011) retrievals. There are significant
differences in the retrieval algorithms of the two products. For example, the a priori $NO_2$ profiles
of the NASA product is based on data from the Global Modeling Initiative (GMI) model with
yearly varying emissions, whereas the a priori $NO_2$ profiles of the DOMINO product is from the
Tracer Model 4 (TM4) without interannual variations in emissions. In addition, for the NASA
product, the stratospheric contribution to the tropospheric column is estimated from the GMI
model simulation. In contrast, for the DOMINO product, the stratospheric contribution is based on
the assimilation of OMI data into the TM4 model.



Starting in 2007, anomalies were found in OMI data and diagnosed as attenuated measured
radiances in certain cross-track positions. This instrument degradation has been referred to as the
"row anomaly". In order to ensure the quality and stability of the data, the following filters are
applied in our analysis for both OMI products (NASA and DOMINO):

1) Tropospheric Column Flag = 0

2) Surface Albedo < 0.3

3) Cloud Radiance Fraction < 0.5

4) No edge data (rows 1-5, 56-60)

5) No row anomaly data (rows 27-55 for the whole period 2005-2015)

After the application of the filters, the number of measurements over the US is about 185,000 per
month in 2010. Thus, we expect the uncertainties in the monthly/annual mean $NO_2$ columns due
to random errors are small. The discrepancy between the two OMI products (see Figures 1a-b) is
mainly caused by the two different retrieval algorithms.
**2.2 AQS and NAPS surface in-situ $NO_2$ concentration**
We use daily-averaged in-situ surface $NO_2$ measurements from the EPA AQS network,
and the Environment Canada NAPS network. The AQS/NAPS networks collect ambient air
pollution data from monitoring stations located in urban, suburban, and rural areas. In the analysis
here, the daily data are averaged to obtain monthly mean concentration at each station.
**2.3 Flash rate density from Lightning Imaging Sensor (LIS)**
LIS is a component of the NASA Tropical Rain Measuring Mission (TRMM). It measures
total optical pulses from cloud-to-ground and intracloud lightning flashes during both day and
night with global coverage (42.5°S-42.5°N) in the period 1995-2014. Monthly flash rate density
(flash/km2) with 2.5°x2.5° resolution is used in this work (Cecil et al. 2006).



**2.4 NOAA Niño 3.4 index**


The Niño 3.4 index comprises sea surface temperature averaged across the region (5°S–

5°N, 170°W–120°W), and their monthly anomalies relative to the 1982–2015 means to constitute
the indices. Years with positive values (>0.5) are considered as El Niño, whereas years with
negative values (< -0.5) are considered as La Niña.
**2.5 Passive tracer simulation using GEOS-Chem model**

The GEOS-Chem global chemical transport model (CTM) [www.geos-chem.org] is driven

by assimilated meteorological fields (MERRA) from the NASA Goddard Earth Observing System
at the Global Modeling and data Assimilation Office. We use version v9-01-03 of GEOS-Chem at
a horizontal resolution of 4°x5°. Bertram et al. (2013) indicated the dominant role of long-lived
reservoir species in the transpacific transport of reactive nitrogen using aircraft measurements from
the INTEX-B campaign. Although the lifetime of tropospheric $NO_x$ is short, the lifetime of long-
lived reservoir species is much longer, for example, the lifetime of free tropospheric PAN is about
1 month. In order to assess the effects of physical transport processes on the long-range transport
of reactive nitrogen, we performed a "passive" tracer simulation, with a constant and uniform
timescale for loss of 15 days (i.e. 360 hours) over the period of 2005-2015 following the approach
of Jiang et al. (2016b). The global a priori surface $NO_x$ emissions (anthropogenic, biomass burning
and soil emissions) are fixed at 2005 level. For each time step (one hour), the tropospheric $NO_2$ is
calculated by: $NO_2^t = NO_2^{t-1} e^{-1/360}$. The lightning $NO_x$ emissions are not included in the
simulation. The 15-day lifetime was selected to provide an approximation for the variation of free
tropospheric $NO_x$ via the formation and transport of long-lived reservoir species, due to changes
in meteorology. Although actual lifetimes of long-lived reservoir species will vary, we found that
15-days was a reasonable compromise to understand the influence of decadal-scale variability on



long-range transport patterns.
**3. Results and Discussion**
Figures 1a-b show the variations of mean tropospheric $NO_2$ columns from OMI (NASA
and DOMINO products) over the US and east China, respectively. Although there is a significant
bias in the magnitude of tropospheric $NO_2$ column between two OMI products, indicating the
influence of different retrieval algorithms, this bias should not affect the trend analysis, as
demonstrated by the consistent interannual variations between the two data products. Figure 1c
shows percent changes, relative to 2009, of the annual mean tropospheric $NO_2$ columns over the
US, and of the total US $NO_x$ emissions (anthropogenic + biomass burning) from the US EPA
(https://www.epa.gov/air-emissions-inventories/air-pollutant-emissions-trends-data).    There    is
good agreement between the changes in the OMI retrievals and the emissions estimates in the
2005-2009 period: the annual slope of the EPA's estimates is -6.4%±0.03% (slope of linear
regression ± uncertainty of slope), and the annual slopes of the two sets of OMI retrievals are -
6.8%±1.1% (NASA) and -8.0%±0.8% (DOMINO). Conversely, we find a large, growing
separation in the 2009-2015 period: the annual slope of the EPA's estimates is -4.6%±0.03%,
whereas the annual slopes of OMI retrievals are -0.5%±0.6% (NASA) and 1.6%±1.1%
(DOMINO). Figures 1d-e show the percent changes in the seasonal mean tropospheric $NO_2$
columns from OMI retrievals, and in the EPA's estimates (annual mean). The divergence between
the seasonal $NO_2$ columns and the emissions is similar to that shown in Figure 1c, suggesting there
is no obvious seasonal dependence.
Our intention here is to understand the possible reasons for the divergence between
observed changes in $NO_2$ vs. changes expected from $NO_x$ emissions. Figure 2 depicts the potential
hypotheses that could explain the divergence:



• Hypothesis 1 (H1): Increasing local $NO_x$ emissions missing from the EPA inventories
• Hypothesis 2 (H2): Time dependent OMI retrieval errors
• Hypothesis 3 (H3): Non-local sources
**3.1 Increasing local $NO_x$ emissions (H1)**
Figure 3a shows the differences of mean surface $NO_2$ concentrations, as measured by the
AQS and NAPS network, from 2009-2010 to 2014-2015. These time periods were chosen to
determine changes in surface $NO_2$ concentrations over the period of 2009-2015 with sufficient
stattistics. Figure 3b shows the same variations averaged with 4°x5° resolution (i.e., at the GEOS-
Chem grid points). The surface stations demonstrate dramatic decreases of surface $NO_2$
concentrations in the period of 2009-2015. The consistent decreasing trends in the surface $NO_2$
concentrations and the EPA's emission estimates during the period of 2009-2015 suggests that the
divergence between the OMI retrievals and the EPA's emission estimates is not caused by US
local emissions. We note that our analysis based on surface in-situ measurements may not provide
sufficient representation for emissions from oil and gas exploration and production. However,
though potentially important locally, these activities only contribute about 5% to total US $NO_x$
emissions based on EPA's estimates. Therefore, we do not expect significant contributions to the
overall changes in tropospheric $NO_2$ from these sources.
Similarly, we expect limited contributions from other sources which are not included in
EPA's inventory. The contribution from aircraft $NO_x$ is only about 3% of total US $NO_x$ emissions
(Skowron et al. 2014). Soil $NO_x$ emissions account for up to 40% of the tropospheric $NO_2$ columns
in summer over US rural areas (Hudman et al. 2010), but only contribute a few percent of the US
annual mean tropospheric $NO_2$ column. Similarly, $NO_x$ production by lightning is stronger in
summer, with an estimated annual contribution of 15% to the total emissions. Because the



discrepancies between the OMI retrieveals and the EPA's emission estimates lack a clear seasonal
dependence (see Figures 1d-e), we expect negligible contributions from soil and lightning $NO_x$
emissions to the enhanced tropospheric $NO_2$ in the period of 2009-2015. Furthermore, Figure 4
indicates that the flash rate density over North America from LIS is uncorrelated with the observed
$NO_2$ variation.

**223    3.2 Time dependent OMI retrieval errors (H2)**

The quality of the OMI retrievals has been evaluated with surface in-situ measurements.
Lamsal et al. (2015) reported that the correlation between OMI $NO_2$ tropospheric columns (NASA)
and the AQS surface in-situ $NO_2$ measurements was 0.68 for the period 2005-2010. Hoek et al.
(2015) indicated that the correlation between the OMI $NO_2$ tropospheric columns (DOMINO) and
surface in-situ measurements in the Netherlands was 0.74 at 2007. The stability of our analysis
based on OMI retrievals (NASA and DOMINO) is ensured by the strict quality filters; these ensure
that changes in OMI sampling due to detector problems (e.g. row anomaly) do not affect our
conclusions.
Figure 5 shows the annual slopes of tropospheric $NO_2$ columns from OMI (NASA and
DOMINO) over the period of 2005-2015. Both OMI products (NASA and DOMINO), with
various a priori models (GMI and TM4) and algorithms, show consistent variations over the
northern Pacific Ocean and the western US: insignificant changes in the period 2005-2008 (Figure
5a-b), positive changes in the period 2009-2012 (Figure 5c-d) and insignificant changes in the
period 2013-2015 (Figure 5e-f). Using the Berkeley High-Resolution (BEHR) $NO_2$ product for
OMI, Russell et al. (2012) obtained similar positive change over the western US with the Weather
Research Forecasting Chemistry model (WRF-Chem) as an a priori model in the period 2005-
2011. The consistency among the various data products suggests that the variations in the retrieved



OMI $NO_2$ over the period of 2005-2015 are not caused by systematic biases.
While our adherence to published data quality filters should ensure the observed OMI
based $NO_2$ values are robust, we note as a caveat that we cannot unequivocally confirm these
changes with independent data due to a lack of either total column or free-tropospheric $NO_2$
measurements over the observed region. For example, Mt. Bachelor Observatory (MBO) station
has free troposheric $NO_2$ observations in the period of 2005-2009 and the CARIBIC aircraft
measurements only provide free tropospheric $NO_2$ observations over the US since 2014. Using
remotely sensed $NO_2$ measurements from the Global Ozone Monitoring Experiment-2 (GOME-
2), Miyazaki et al. (2017) found a slight increase of tropospheric $NO_2$ columns over the US in the
period of 2009-2012, consistent with our result. However, a significant sudden decrease of
retrieved tropospheric $NO_2$ columns from GOME-2 has been observed since July 2013, associated
with the change in the measurement mode (http://projects.knmi.nl/atcom/ news.php?id=44).

**3.3 Non-local sources (H3)**

We have demonstrated that hypotheses H1 and H2 are not likely the dominant factors,
which leaves hypothesis H3 (non-local sources) as possible important contributors. Figures 5g-h
show the annual slope of tropospheric $NO_2$ columns (percent base) from OMI (NASA and
DOMINO) over the period of 2009-2015. Our analysis demonstrates a significant positive change
over the northern Pacific Ocean during this time period, and an insignificant but positive change
over the western US, suggesting possible contributions from transpacific transport to tropospheric
$NO_2$ over the western US. Over the period of 2005-2008, the lack of change in tropospheric $NO_2$
columns over the northern Pacific Ocean (Figure 5a-b) indicates the dominant role of local sources
to the decrease of US tropospheric $NO_2$ in this period. Conversely, the increase in tropospheric
$NO_2$ columns over northern Pacific Ocean over the period of 2009-2012 (Figure 5c-d) is consistent





with the appearance of a discrepancy between the OMI retrievals and EPA's emission estimates
(Figure 1c). Accompanying with the observed decrease of Chinese $NO_2$ emissions (Figure 1b), no
significant change is observed over the northern Pacific Ocean over the period of 2013-2015
(Figure 5e-f).

Decadal climate variability has non-negligible influences on tropospheric compositions by

affecting the physical and chemical processes. For example, Lin et al. (2014) indicated that
transpacific transport of $O_3$ is modulated by decadal variability of El Niño–Southern Oscillation
(ENSO). El Niño is defined as the appearance of anomalously warm water off northern Peru and
Ecuador in December. The atmospheric component tied to El Niño is called the Southern
Oscillation (Trenberth 1997). To the best of our understanding, ENSO is the dominant climate
phenomenon linked to extreme weather conditions globally (Cai et al. 2015), and it also exerts a
major influence on the interannual variability of $O_3$ in the troposphere (Doherty et al., 2006).
Following Jiang et al. (2016b), we conducted an analysis using an idealized passive tracer to assess
the possible influences of transport patterns. We performed a GEOS-Chem model simulation for
tropospheric $NO_2$ over the period of 2005-2015 with an $NO_2$-like tracer with a constant 15-day
lifetime and fixed (2005 level) surface $NO_x$ emissions. The passive tracer simulation with constant
lifetime avoids the possible influences from uncertainties in the modeled nonlinear $NO_x$ chemistry,
particularly, the conversion between $NO_x$ and its longer-lived reservoir species.

Figures 6a-c show that even with emissions held constant, interannual variations in

transport produce differences in $NO_2$ (or the passive tracer) columns over the eastern Pacific.
Based on the passive tracer simulation, transpacific transport decreased over the period of 2005-
2008 (Figure 5a). During this four-year period, declining transport efficiency appears to have offset
the increase of Asian emissions, resulting in insignificant changes of tropospheric $NO_2$ columns



over the northern Pacific Ocean (Figures 5a-b), and consequently, good agreement between the
OMI retrievals and the EPA's emission estimates (Figure 1c). The efficiency of transpacific
transport is more stable over the period of 2009-2012 (Figure 6b), which allows stronger
transpacific transport of rising Asian emissions. It leads to positive changes of tropospheric NO$_2$
columns over the northern Pacific Ocean (Figure 5c-d), and consequent growing discrepancy
between the OMI retrievals and the EPA's emission estimates (Figure 1c). Increasing efficiency
in transpacific transport over the period of 2013-2015 (Figure 6c) counteracts the decrease of
Chinese NO$_2$ (Figure 1b), again resulting in no change in the tropospheric NO$_2$ column over
northern Pacific Ocean (Figure 5e-f) and relatively flat changes in US tropospheric NO$_2$ column.
Figure 6d shows the comparison between regional mean of passive tracer columns over the
northern Pacific Ocean with the NOAA Niño 3.4 index. There is strong correlation between
transpacific transport and ENSO: the transpacific transport is stronger in El Niño years and weaker
in La Niña years, demonstrating strong influence of decadal climate variability on the transpacific
transport.
Brown-Steiner et al. (2011) showed that Asian O$_3$ is transported across Pacific Ocean
primarily in the lower/middle troposphere in winter and spring, and primarily in the middle/upper
troposphere in summer and fall, but that differences in tropospheric column remain small for
different seasons. Figure 7 shows the seasonal mean tropospheric columns for the NO$_2$-like tracer
between 2005-2015. Our analysis indicates the transpacific transport, with constant 15-day lifetime,
is strongest in summer and spring. However, the lifetime of PAN approximately doubles for every
4°C decrease in temperature. Consequently, due to the temperature effect associated with the
change in transport pathways from the lower troposphere (in spring) to the upper troposphere (in
fall), transport in fall, relative to that in spring, should be greater than that shown in Figure 7c.



Similarly, the actual transpacific transport of NO$_2$ in winter, relative to that in spring, should be
greater than shown in Figure 7d because of the temperature differences between winter and spring.
Thus, in agreement with Brown-Steiner et al. (2011), our analysis would suggest weak seasonality
of transpacific transport of reactive nitrogen, consistent with the observed changes in OMI NO$_2$
over the US (Figures 1d-e). It should be noticed that our conclusion about weak seasonality of
transpacific transport is different from previous studies (e.g. Liang et al. 2004) based on carbon
monoxide (CO) simulations. The discrepancy in the seasonality is associated with the strong
springtime biomass burning CO emissions in southeast and boreal Asia, whereas the contribution
of biomass burning to tropospheric NO$_x$ is relatively small.
**4. Conclusions**

In this work, we investigated the variation of US tropospheric NO$_2$ in the past decade, to

evaluate the contribution of non-local sources to the tropospheric NO$_x$ budget. We demonstrated
significant divergence between the time variation in tropospheric NO$_2$ columns from the OMI
retrievals and the EPA's NO$_x$ emission estimates. Our analysis suggests limited contributions from
local effects such as fossil fuel emissions, lightning, or instrument artifacts, and indicates possible
important contributions from long-range transport of Asian emissions that are modulated by ENSO.
Passive tracer simulation with fixed emissions demonstrates that the intensity of transpacific
transport is stronger in El Niño years and weaker in La Niña years. The unexpected important
contributions from long-range transport contradict assumptions of weak long-range transport for
NO$_x$, suggesting potential underestimation of transported reactive nitrogen in the state of the art
models.

In related studies, long-term free tropospheric O$_3$ observations over Europe demonstrated

a significant increase in the past three decades (e.g. Logan et al. 2012; Parrish et al. 2012), whereas





state of the art models cannot reproduce this variation (Parrish et al. 2014). Enhanced long-range
transport of $NO_x$ could potentially reconcile the large discrepancy between modeled and observed
free tropospheric $O_3$ over Europe. Because of uncertain processes in the long-range transport of
reactive nitrogen (Ye et al. 2016) and the dominant role of long lifetime reservoirs (Bertram et al.
2013), we are not able to quantify the different contributions to the observed tropospheric $NO_x$ in
this study. This quantification will require comprehensive observations and modeling efforts to
understand the formation and transport of long lifetime reservoirs of reactive nitrogen.

**Acknowledgments**: We acknowledge the OMI tropospheric $NO_2$ column data from www.temis.nl
and disc.sci.gsfc.nasa.gov. We acknowledge the flash rate density data from ghrc.nsstc.nasa.gov.
We thank the Environmental Protection Agency for providing their national $NO_x$ emission data
and surface in-situ $NO_2$ measurements. We thank the Environment Canada for providing their
surface in-situ $NO_2$ measurements. We thank Lee Murray for useful discussions. The National
Center for Atmospheric Research (NCAR) is sponsored by the National Science Foundation. Part
of this work was carried out at the Jet Propulsion Laboratory, California Institute of Technology,
under a contract with the National Aeronautics and Space Administration. Support for Emily V.
Fischer and Liye Zhu was provided by NASA Award Number NNX14AF14G. Support for Daven
K. Henze was provided by NASA HAQAST.

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





**Tables and Figures**
**Figure 1.** (a-b): monthly mean (dash lines) and annual mean (solid lines) tropospheric $NO_2$ column
over contiguous United States and East China from OMI (NASA and DOMINO) products; (c-e):
percent changes of annual mean (c) and seasonal mean (d-e) tropospheric $NO_2$ column over the
US from the OMI and EPA's emission estimates, normalized at 2009.
**Figure 2.** Schematic figure showing the sources of tropospheric $NO_x$.
**Figure 3.** (a) difference of mean $NO_2$ concentrations of surface in-situ measurements (AQS and
NAPS stations) from 2009-2010 to 2014-2015; Blue (red) means decrease (increase) of $NO_2$
concentrations. (b) same as panel a, but averaged with 4°x5° resolution.
**Figure 4.** Flash rate density ($1x10^{-3}$ flash/$km^2$/month) over North America (15°N-42.5°N, 130°W-
60°W) from Lightning Imaging Sensor (LIS).
**Figure 5.** (a-f) annual slope of tropospheric $NO_2$ column (unit $1x10^{15}$ molec/$cm^2$) from OMI
(NASA and DOMINO products); (g-h) same as panels a-f, with percent (%) as unit.
**Figure 6.** (a-c) Annual slope of passive tracer column (percent base). The passive tracer is
simulated with GEOS-Chem model with constant 15-day lifetime. The surface $NO_x$ emissions are
fixed at 2005 level. The lightning $NO_x$ emissions are not included in the simulation. (d) Blue line:
regional mean (box in panel a) of passive tracer column, normalized by the 11-year mean (2005-
2015). The black line shows the NOAA Niño 3.4 index.
**Figure 7.** Seasonal mean passive tracer column (2005-2015), with $10^{16}$ molec $cm^{-2}$. The passive
tracer is simulated with GEOS-Chem model with constant 15-day lifetime. The surface $NO_x$
emissions are fixed at 2005 level. It should be noticed that the actual transpacific transport of
reactive nitrogen in fall and winter is stronger than panels c-d due to the decrease of temperature.





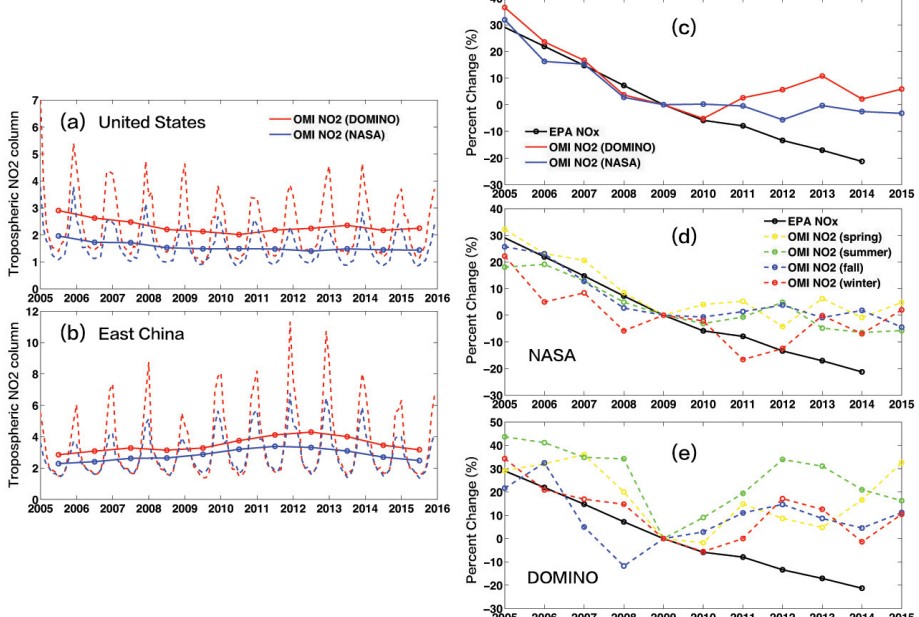


**Figure 1.** (a-b): monthly mean (dash lines) and annual mean (solid lines) tropospheric $NO_2$
column over contiguous United States and East China from OMI (NASA and DOMINO)
products; (c-e): percent changes of annual mean (c) and seasonal mean (d-e) tropospheric $NO_2$
column over the US from the OMI and EPA's emission estimates, normalized at 2009.


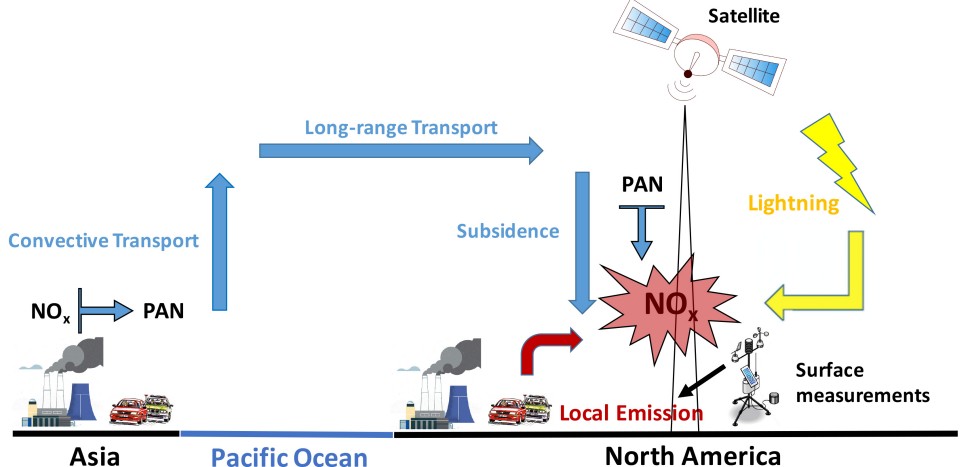


**Figure 2.** Schematic figure showing the sources of tropospheric $NO_x$.






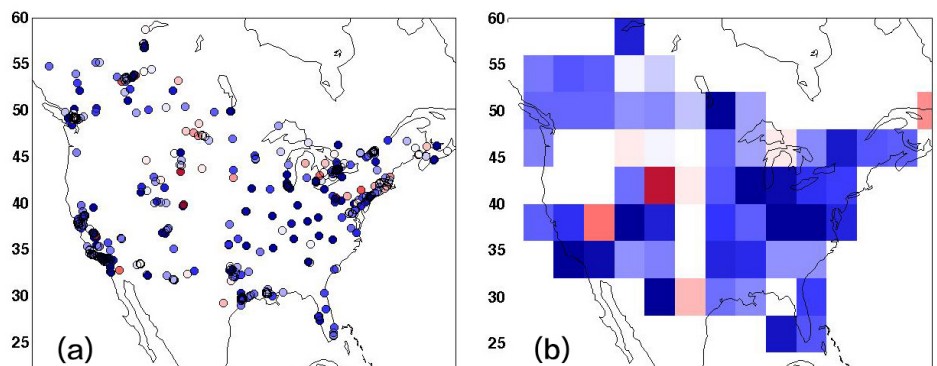

**Figure 3.** (a) difference of mean $NO_2$ concentrations of surface in-situ measurements (AQS
and NAPS stations) from 2009-2010 to 2014-2015; Blue (red) means decrease (increase) of
$NO_2$ concentrations. (b) same as panel a, but averaged with 4°x5° resolution.

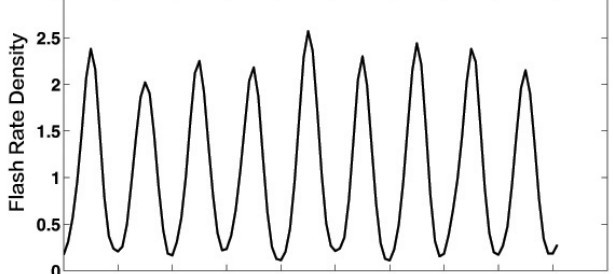

**Figure 4.** Flash rate density ($1x10^{-3}$ flash/$km^2$/month) over North America (15°N-42.5°N,
130°W-60°W) from Lightning Imaging Sensor (LIS).






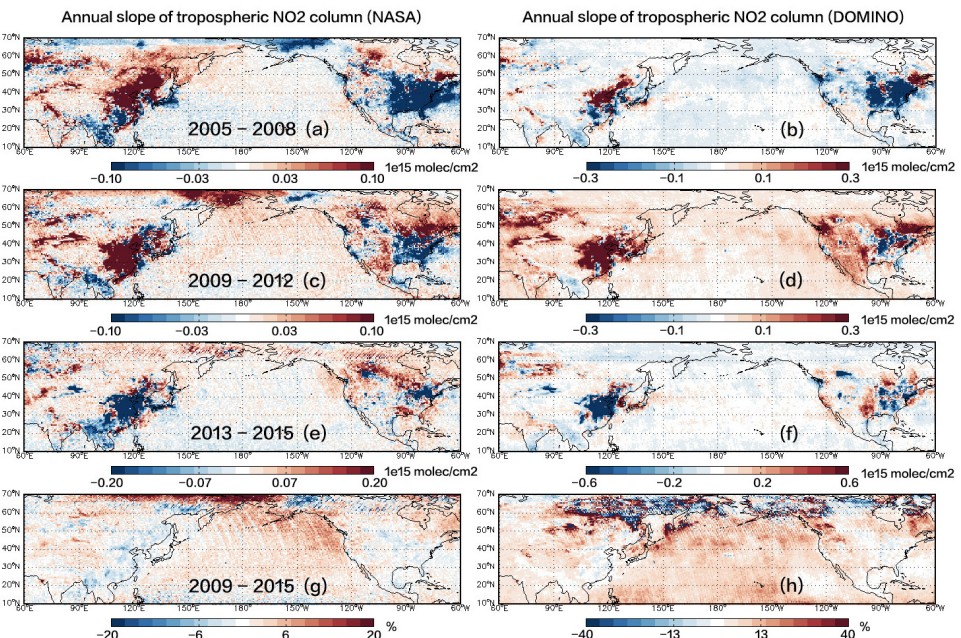


**Figure 5.** (a-f) annual slope of tropospheric NO$_2$ column (unit $1\times10^{15}$ molec/cm$^2$) from OMI (NASA and DOMINO products); (g-h) same as panels a-f, with percent (%) as unit.







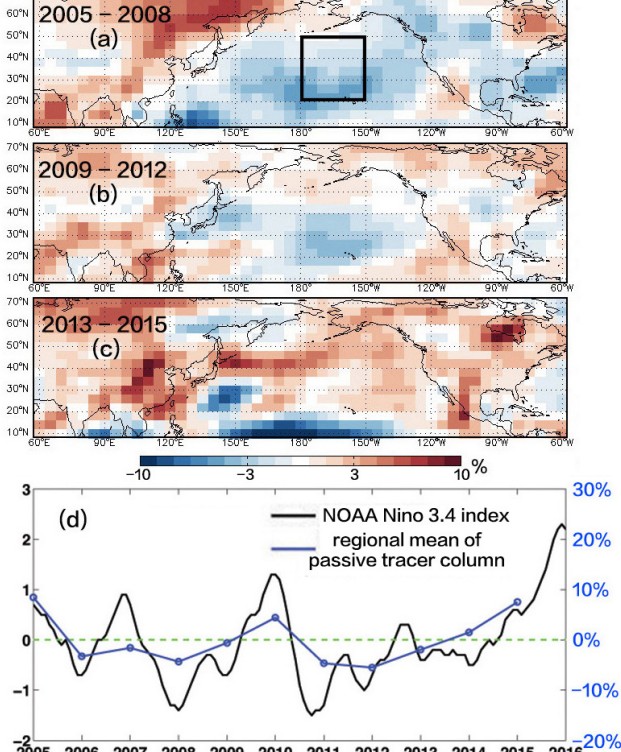

**Figure 6.** (a-c) Annual slope of passive tracer column (percent base). The passive tracer is
simulated with GEOS-Chem model with constant 15-day lifetime. The surface $NO_x$ emissions
are fixed at 2005 level. The lightning $NO_x$ emissions are not included in the simulation. (d)
Blue line: regional mean (box in panel a) of passive tracer column, normalized by the 11-year
mean (2005-2015). The black line shows the NOAA Niño 3.4 index.
592



593

**Figure 7.** Seasonal mean passive tracer column (2005-2015), with $10^{16}$ molec cm$^{-2}$. The passive
tracer is simulated with GEOS-Chem model with constant 15-day lifetime. The surface NO$_x$
emissions are fixed at 2005 level. It should be noticed that the actual transpacific transport of
reactive nitrogen in fall and winter is stronger than panels c-d due to the decrease of temperature.

598