# Peer review of "nitrogen oxides over United States Zhe Jiang1, Helen Worden1, John R. Worden2, Daven K. Henze3, Dylan B. A. Jones4, Avelino F. Arellano5, Emily V. Fischer6, Liye Zhu6, Kazuyuki Miyazaki2,7, K. Fol"

_Atmospheric Chemistry and Physics, 2017_

## Referee Comment (RC1) · Anonymous Referee #2 · 27 Jun 2017

Review of "Inconsistent decadal variations between surface and free tropospheric nitrogen oxides over United States" by Y. Zhe Jiang et al.

MS Number: acp-2017-382

**Summary:**

This paper analyzes satellite measurements of tropospheric $NO_2$ columns and compares their temporal trends with those estimated for U.S. NOx emissions. Based on discrepancies between these time trends the authors reach conclusions of rather large significance: "The unexpected important contributions from long-range transport contradict assumptions of weak long-range transport for NOx, suggesting potential underestimation of transported reactive nitrogen in the state of the art models." In my opinion papers reporting unexpected findings that have large significance must be supported with rock-solid, rigorous analysis, including robust statistical confidence limits; such analysis is missing from this paper. Indeed the authors may well be analyzing statistically insignificant noise in the satellite record. I suggest that this paper be rejected. If the authors can greatly improve the analysis, and reach the same or similar conclusions, with the necessary statistical analysis, then they should resubmit the paper. The following comments describe my major concerns.

**Major issues:**

1) My primary concern regarding this paper is a lack of rigorous analysis of confidence limits. Let's imagine a time series of satellite measurements of annual mean and seasonal mean tropospheric $NO_2$ column over the US, and that the quantitation limit of that column was equal to the $NO_2$ column in 2009. It that case before 2009 we would be able to discern decreasing $NO_2$ columns larger than that in 2009, but after 2009 we would be only able to discern noise, regardless of any continuing decrease in the actual $NO_2$ column. To my eye, that pretty well describes the situation in Figure 1c. Now, if this imaginary satellite measurement record is analyzed on a seasonal basis, the noise would be larger, and the decrease would disappear into the noise even earlier than 2009. Again, to my eye, that pretty well describes the situation in Figures 1d and 1e. A rigorous analysis that clearly demonstrates a statistically significant (at the 2 sigma or 95% confidence level) divergence of surface and satellite nitrogen oxide measurements must be given before this paper can be considered for publication. My fear is that this entire paper is simply analyzing statistically insignificant differences between the time trends extracted from the satellite data and the NOx emission estimates.

2) I am not an expert in satellite data by any means, but if I understand correctly, the satellite most accurately and precisely determines the total $NO_2$ column. To derive the tropospheric column, the stratospheric column must be subtracted. Over remote and even rural areas, the stratospheric contribution is a significant fraction of the total. The statistical analysis discussed in the preceding point must fully consider the systematic and random errors that inevitably arise from this subtraction.

3) The meaning of the black curves in Figs. 1c-1e must be clarified. The figure caption and much of the discussion suggest that these are EPA emissions estimates. If this is true, the source of these estimates and their spatial extent (continental U.S.?) must be made clear. Section 2.2 discusses in-situ surface $NO_2$ measurements from the EPA AQS network, and the Environment Canada NAPS network. It would be helpful to compare trends derived from

these $NO_2$ measurements with the emission estimates, and to demonstrate that the trends are statistically consistent. The title of the paper is "Inconsistent decadal variations between surface and free tropospheric nitrogen oxides over United States"; this implies to my mind that Figure 1 should be showing surface nitrogen oxide measurements, not nitrogen oxide emission estimates. Emissions estimates have large errors and these errors vary from year-to-year.

4) Lines 204-208 mention consistent decreasing trends in the surface $NO_2$ concentrations and the EPA's emission estimates. However, no statistical analysis is presented to show that these trends are consistent; a rigorous statistical analysis is required.

5) Lines 217-220 state "the discrepancies between the OMI retrieveals and the EPA's emission estimates lack a clear seasonal dependence (see Figures 1d-e)." The seasonal average OMI retrievals are so noisy that a quite significant seasonal dependence could be present, but could not be discerned in those figures. If this discussion is retained, a rigorous statistical analysis is again required.

6) Section 3.2 is a qualitative discussion of the relative changes in tropospheric $NO_2$ column obtained from various satellite retrievals. Without a robust, quantitative, rigorous statistical analysis, this discussion is of little value. This section must be improved or eliminated.

7) Section 3.3 discusses Figure 5 in detail. Figures 5g and 5h show little trends over the 2009 to 2015 period, which seems to support the authors' hypothesis. However, Figures 5c-f show trends for shorter periods, and show strong, physically unreasonable trends. For example, large increases over the Pacific Northwest-southwest Canada and southeastern Canada in the earlier period are at least partially compensated by significant decreases in the same area in later periods. This reinforces my concern raised in Issue 1 above - trends derived from the satellite record seem to be on the edge of statistical significance.

8) The paragraph on lines 282-300 is a qualitative discussion of decadal climate variability and how it might be affecting $NO_2$ transpacific transport. To be acceptable, this discussion requires strong quantitative support with robust statistical analysis.

9) Lines 297-300 state "There is strong correlation between transpacific transport and ENSO: the transpacific transport is stronger in El Niño years and weaker in La Niña years, demonstrating strong influence of decadal climate variability on the transpacific transport." This statement refers to Figure 6d. The right hand axis only spans a range of ~ +8% to -5%, hardly a "strong influence". The actual strength of the correlation is not given. At least an $r^2$ value should be given.

10) The paragraph on lines 301-318 is a qualitative discussion of a model study of transpacific transport of an inert tracer with a constant lifetime designed to mimic PAN. However, the transport of PAN has a very strong altitude dependence due to its temperature dependent lifetime. In my opinion this simulation is too crude to be relevant.